# Pt Cluster Modified h-BN for Gas Sensing and Adsorption of Dissolved Gases in Transformer Oil: A Density Functional Theory Study

**DOI:** 10.3390/nano9121746

**Published:** 2019-12-08

**Authors:** Yingang Gui, Tao Li, Xin He, Zhuyu Ding, Pingan Yang

**Affiliations:** 1College of Engineering and Technology, Southwest University, Chongqing 400715, China; litao199811@outlook.com (T.L.); lovelife824@163.com (X.H.); dingzy@swu.edu.cn (Z.D.); 2Key Laboratory of Industrial Internet of Things & Networked Control, Ministry of Education, Chongqing University of Posts and Telecommunications, Chongqing 400065, China; yangpa@cqupt.edu.cn

**Keywords:** Pt cluster modified h-BN, adsorption and sensing, oil dissolved gases, simulation

## Abstract

Hexagonal-Boron nitride nanotubes (h-BN) decorated with transition metals have been widely studied due to their enhanced physicochemical properties. In this paper, Pt cluster-modified h-BN is proposed as a sensitive material for a novel gas sensor for the online malfunction monitoring of oil-immersed transformers. The inner oil is ultimately decomposed to various gases during the long-term use of oil-immersed transformers. Exposure to excessively high temperatures produces the alkanes CH_4_ and C_2_H_6_, whereas different degrees of discharge generate H_2_ and C_2_H_2_. Therefore, the identification of H_2_, CH_4,_ and C_2_H_2_ gas efficiently measures the quality of transformers. Based on the density functional theory, the most stable h-BN doped with 1–4 Pt atoms is employed to simulate its adsorption performance and response behavior to these typical gases. The adsorption energy, charge transfer, total density of states, projected density of states, and orbital theory of these adsorption systems are analyzed and the results show high consistency. The adsorption ability for these decomposition components are ordered as follows: C_2_H_2_ > H_2_ > CH_4_. Pt cluster-modified h-BN shows good sensitivity to C_2_H_2_, H_2_, with decreasing conductivity in each system, but is insensitive to CH_4_ due to its weak physical sorption. The conductivity change of Pt_n_-h-BN is considerably larger upon H_2_ than that upon C_2_H_2_, but is negligible upon CH_4_. Our calculations suggest that Pt cluster modified h-BN can be employed in transformers to estimate their operation status.

## 1. Introduction

Oil-immersed transformers are extensively employed in modern power systems. Oil-immersed transformers are recognized as one of the most essential electric equipment because of their outstanding power conversion function and universal application. However, inevitable malfunctions associated with the long-term use of transformers, such as local overheating, partial discharge, and equipment wear out, are likely to cause catastrophic damage to the entire power system. Thus, effective methods for the online monitoring of transformers have been developed by the power sector. Along with different types of faults, various impurity gases, including H_2_, CH_4_, C_2_H_6_, and C_2_H_2_, are generated by transformers [1,2,3]. Exposure to excessive temperature produces the alkanes CH_4_ and C_2_H_6_, whereas different degrees of discharge generate H_2_ and C_2_H_2_. Given that H_2_, CH_4_, and C_2_H_2_ are the three typical gases in transformer faults, their identification effectively measures the quality of transformers. The condition of the transformer can be speculated by analyzing the concentration of these three typical gases and the variation of conductivity in the oil. Given the good sensitivity and versatility of gas sensors comparing with other testing methods [4], a new type of gas sensor with selectivity and sensitivity can be developed to strengthen the running stability of oil-immersed transformers.

Boron nitride nanotubes (BN), also known as white graphene, are widely employed in various fields. BN exhibits excellent performance, including heat resistance and acid-alkali resistance [5,6]. BN is an insulator with good gas adsorption performance due to its large specific surface area. BN is employed in biological and chemical fields. The mechanism of action between metal-doped BN and ozone, and the adsorption behavior of the drug metformin on BN fullerenes have been studied [7,8]. Hexagonal-BN nanotubes (h-BN) (8,0) are widely applied in gas adsorption and sensing [9,10,11,12]. Nevertheless, intrinsic h-BN exhibits a limited response to some inert gases, such as CH_4_ and C_2_H_2._ Metal cluster modification, such as Pd, Pt, and Ni doping [2,13,14], is commonly employed to strengthen gas sensitive responsiveness. Pt is one of the most widely used modification metals and is considered in this work.

Metal-doped material on gas sensing has developed rapidly, with numerous transition metals considered. We proposed the use of Pt cluster-modified h-BN as a promising sensing material for the detection of dissolved gases in oil-immersed transformers in order to monitor the performance of transformers. Given the activity and catalytic properties of Pt, Pt cluster modification can ameliorate the chemical activity of the h-BN surface so obtaining improved adsorption. This doped structure has rarely been studied. Thus, we analyzed the best adsorption and structural stability of Pt cluster-modified h-BN doped with 1–4 Pt atoms. We then selected the optimal structures to examine the adsorption mechanism of H_2_, CH_4_, and C_2_H_2_. The mechanisms of how the Pt cluster-modified h-BN reacts with these components must be determined to allow for the preparation of sensors to detect transformer malfunctions.

## 2. Computational Details

The entire computations in this work were conducted based on the density functional theory (DFT) [15,16]. Spin-polarized calculations were unrestricted. The supercell periodic boundary condition was set as 20 Å × 20 Å × 8.5 Å to prevent interaction between neighboring cells [17,18]. The generalized gradient approximation was utilized for computation by Perdew Burke Ernzerhof for the approximation treatment of exchange-correlation functional, which is widely used in the calculation of extensive materials and their surfaces [19,20,21]. The Brillouin-zone was carried out by the Monkhorst-Pack scheme with the k-point set to 3 × 3 × 1, which presents good approximation for h-BN [22,23]. The double numerical plus polarization was selected as the atomic orbital basis set, whereas the DFT semi-core pseudopotential method was applied considering the relativistic effect of transition elements [24]. The energy tolerance accuracy, maximum force, and displacement were selected as 10^−5^ Ha, 2 × 10^−3^ Ha/Å, and 5 × 10^−3^ Å, respectively [25,26]. For the stationary electronic structure, a precise convergence criterion of 1.0 × 10^−6^ Ha for self-consistent field tolerance was employed [27].

We optimized the geometry of the H_2_, C_2_H_2_, and CH_4_ in oil, as well as the majorization treatment of the intrinsic h-BN structure. Pt cluster modification was considered to obtain improved adsorption performance. In this work, all possible Pt cluster-modified h-BNs doped with 1–4 Pt atoms were optimized. The best performance and the most stable structure with different Pt cluster decorations was obtained by calculating the adsorption energy (*E*_d_) of doped fabric, and is defined in Equation (1), combined with the specific bond length and collection shape.
*E*_d_ = *E_Ptn/h-BN_* − *E_h-BN_* − *E_Ptn_*(1)
where *n* represents the number of doping atoms and *E*_d_ expresses the energy released during binding. To analyze the mechanism and effect of the adsorption process, we calculated the charge transfer (*Q*_t_) and adsorption energy (*E*_ads_) based on Equations (2) and (3).
*Q*_t_ = *Q*_1_ − *Q*_2_(2)
*E*_ads_ = *E*_gas/suf_ − *E*_gas_ − *E*_suf_(3)
where *Q*_t_ indicates the charge transfer amount from the gas molecules to the surface of Pt cluster modified h-BN though Mulliken population analysis while *Q*_1_ and *Q*_2_ represent the total net charge after and before gas adsorption, respectively. In Equation (3), *E*_ads_ represents the total energy of the gas molecules before adsorption. A negative *E*_ads_ indicates that the adsorption process is exothermic and spontaneous. The adsorption mechanism can greatly affect the conductivity and gas concentration of the entire system, which is beneficial to fabricate a gas sensor with selectivity and sensitivity. The electrical conductivity is related to energy gap and the change of energy gap will affect conductivity exponentially [28]. To further obtain the conductivity of the entire system, we determined the energy gap (*E*_g_) of the molecular orbital between the highest occupied molecular orbital (HOMO) and the lowest occupied molecular orbital (LUMO), as defined in Equation (4) [29].
*E*_g_ = |*E*_LUMO_ − *E*_HOMO_|(4)

Total density of states (TDOS) and projected density of states (PDOS) in the adsorption are analyzed in detail for comprehension of the mechanism of the adsorption process.

## 3. Results and Discussions

### 3.1. Optimized Structure of Gas Molecules and Intrinsic h-BN

As shown in Figure 1, the optimized structure of gas molecules and intrinsic h-BN are presented. Both H_2_ and C_2_H_2_ are linear gas molecules, whereas CH_4_ is a three-dimensional tetrahedral structure, the H–H bond length in the H_2_ molecule is 0.75 Å, given the weak atomic force and short atomic radius of the H atom. C–H bond lengths are 1.10 and 1.07 Å in CH_4_ and C_2_H_2_ molecules. Different orbital hybridization causes slight difference in the C-H bond lengths. In the CH_4_ molecule, the H–C–H bond angle is 103.39°. The symmetrical structure and four C-H bonds of the CH_4_ molecule imparts its chemical stability. An h-BN with a perfect crystal structure is shown in Figure 1b. The mesh-like hexagonal structure is beneficial to the adsorption performance. The B-N bond length of 1.44 Å is appropriate for structural stability, both axial and circumferential distances are around this value. When all aspects are considered, h-BN is a suitable material for gas adsorption.

### 3.2. Pt Cluster Modified h-BN

Figure 2 shows the different doping structures with 1–4 Pt atoms. The adsorption distance, charge transfer, and adsorption energy are calculated to find the most stable structures with different doping Pt atoms.

Figure 2a shows the most stable structure doped with one Pt atom, which is also the only possible structure at this situation. The atomic arrangement of h-BN exhibits central symmetry and requires the location of the Pt atom to be centered. We considered different situations in the experiments. However, only the bridge site for Pt atom doping can be optimized successfully. The Pt–N bond length is 2.00 Å, which is slightly shorter than the Pt-B bond length (2.19 Å). This phenomenon illustrates the strong force between Pt and N atoms. The adsorption energy (*E*_d_) of −1.98 eV demonstrates fine stability from the perspective of energy.

Two Pt-atom doping is also considered in this work. Different systems have been computed, as presented in Figure 2b1,b2. The two structures are the adjacent and opposite sides of the hexagon where Pt atom doping occurs. The Pt–Pt bond lengths are 2.51 and 2.47 Å, meeting the bond length between heavy metals. The *E*_d_ values of the adjacent and opposite sides are −4.83 and −4.84 eV, as shown in Figure 2b1. Dispersed Pt atoms may cause the two- and single-atom-doped structures to be similar.

As shown in Figure 2c1–c5, five possible models were generated. The clustering of the three metal atoms has been widely employed in the field of materials. Thus, the Pt_3_ cluster is treated emphatically in this work to enhance gas sensitivity response. The surface structure of the intrinsic h-BN is greatly altered after Pt_3_ modification. The atom tends to protrude outward. *E*_d_ values from (c1) to (c5) are −9.02, −8.16, −8.31, −8.30, and −8.30 eV, respectively. A high *E*_d_ proves that the structure shown in Figure 2c1 exhibits stronger stability compared with the other structures.

As shown in Figure 2d1–d4, the Pt_4_ cluster is employed for analysis. Many possible structures are also obtained. We finally selected the most stable structure according to the *E*_d_ and geometry structure, as shown in Figure 2d1. The adsorption energy of this structure is −12.59 eV, which is higher than that of any other structure. The three-dimensional feature also imparts stability to this structure.

To gain insight into the diversification in conductivity, we studied the band structure of systems in Figure 2a,b2,c1,d1. This analysis allows for the advanced understanding of the gas adsorption mechanism. The band structure of the systems is drawn as follows:

As shown in Figure 3, the band gaps from Figure 3a–e are 3.73, 1.50, 0.30, 0.0, and 0.38 eV, respectively. The intrinsic h-BN exhibits a huge band gap of 3.73 eV, which explains the difficulty for electrons to jump from the top of the valence band to the bottom of the conduction band. This result is also consistent with h-BN being an insulator. After Pt cluster modification, the band gap in each case is greatly decreased, enhancing electrical conductivity throughout the system. After Pt_3_ doping, the entire band structure is almost continuous, which may be largely due to the doping position.

### 3.3. Adsorption of H_2_, CH_4_, C_2_H_2_ on the Surface of Intrinsic h-BN

The optimization results for the adsorption of H_2_, CH_4_, and C_2_H_2_ on intrinsic h-BN are presented in Figure 4. The geometric optimization results were calculated as the lowest energy structure according to the DFT, that is, relatively thermodynamic stability, so as to determine the adsorption energy and adsorption distance. According to the final analysis results, intrinsic h-BN exhibits low gas sensitivity response to these typical gases. The H_2_, C_2_H_2_, and CH_4_ molecular structures remain unchanged during the adsorption process, and the long adsorption distance and weak adsorption energy suggest a weak physical mechanism between gas molecules and the intrinsic h-BN. The specific parameters in the adsorption process are provided in Table 1 to further explain the adsorption performance.

The distance of the gas molecules to h-BN is very far, ranging from 3.23 Å to 3.63 Å, explaining the weak interaction between the molecules. The adsorption energies of H_2_, CH_4_, and C_2_H_2_ are −0.08, −0.06, and −0.10 eV, respectively. Such small adsorption energy values also correspond to the long adsorption distance. The final simulation data show that the intrinsic h-BN response to these three gases is ordered as follows: C_2_H_2_ > H_2_ > CH_4_. This order seems to be related to the characteristics of the gas molecules.

### 3.4. Adsorption of Gas Molecules on Pt, Pt_2_ Doped h-BN

Figure 5a–f depicts the gas adsorption system on Pt- and the Pt_2_-doped h-BN. The calculation parameters are shown in Table 2. Based on the optimized adsorption structure, Pt doping can improve the gas sensitivity of the entire system, especially for H_2_ and C_2_H_2_, where strong chemical adsorption process occurs after Pt and Pt_2_ doping. However, the adsorption capacity for CH_4_ is insufficient, with merely weak physical and catalytic effects between the molecules. The adsorption capacity based on the simulation results is as follows: C_2_H_2_ > H_2_ > CH_4_, and the effects in all occasions are strengthened.

Figure 5a–d shows that the H–H bond is broken during the adsorption process. Two H atoms are bonded to the Pt atom at a bond length of 1.55 Å, and the adsorption energy in both cases are 1.94 and 1.97 eV. No difference is found between single- and two-atom doping due to the far distance between the H_2_ molecule and the second Pt atom. After Pt atom doping, *E*_ads_ is increased from −0.08 eV to −1.94 eV. The rapid increase of *E*_ads_ with a large *Q*_t_ (0.26 *e*) indicates that the Pt atom can enhance the adsorption activity of the system, thereby producing a strong chemical action with the H_2_ molecule.

The optimization results for CH_4_ adsorption are shown in Figure 5b–e. Compared with intrinsic h-BN, the Pt- and Pt_2_-doped h-BN can considerably improve the adsorption capacity, as proven by the large *E*_ads_ and short adsorption process distance. The bond lengths of the H atom in the CH_4_ molecules to the Pt atom are 1.76 and 2.53 Å, and the molecular structure of CH_4_ remains substantially unchanged. We speculate that the role between the CH_4_ and Pt is biased toward physical adsorption. The structure in Figure 5e shows a weaker effect, and the stability of the doped structure causes the lower adsorption of diatomic doping compared with that of monoatomic doping.

Figure 5c–e demonstrates the large influence between C_2_H_2_ molecule and doped h-BN. In the single Pt atom-doped system, the Pt–B bond is fractured because of the great force, and the structure of C_2_H_2_ is changed from a straight line to a distorted planar structure. In the two Pt atom-doped system, the degree of distortion is evident. *E*_ads_ with magnitudes of 2.60 and 3.4 eV at different situations suggest a complicated process that occurs, and the adsorption effect is improved by a leap.

For improved understanding of the adsorption for manufacturing suitable gas sensors, the total density of states (TDOS), partial density of states (PDOS), and molecular orbitals for different gas adsorption scenarios were analyzed. An accurate sensor applied to the online monitoring of transformers can be implemented by identifying different parameters, including conductivity and *E*_g_.

As shown in Figure 6a–f, the TDOS is changed after gas adsorption, and Figure 6a1–f1 presents the PDOS of corresponding situations. The adsorption of different gases shows varying effects on the TDOS. The conductivity of the entire system can be precisely modified through TDOS analysis. Finally, a sensitive gas sensor can be fabricated by analyzing the resistance value of the system. PDOS was calculated to explore advanced adsorption mechanisms, particularly the mechanism of intermolecular chemical bonding. A combination analysis can provide accurate simulation data applied to gas sensor manufacturing.

In TDOS, both H_2_ and C_2_H_2_ show remarkable changes in TDOS, but the change in CH_4_ is not evident. In a single-atom-doped system, TDOS is approximately equal to 0 eV between the conduction band and the valence band, reflecting the decline in conductivity. However, the occurrence of this phenomenon is minimal in the two-atom-doped system, because two-Pt-atoms doping increases the conductivity of the system. As shown in Figure 1a, TDOS moves to the right as a whole and then drops slightly at the right side of the Fermi level. This result indicates the reduced number of electronic fillings of the conduction band. Furthermore, the conductive property of the structure is decreased, the reason for which can be analyzed from the distribution of PDOS. Considerable orbital hybridization exists in the 5d orbital of Pt and the 1s orbital of H, which is related to the formation of Pt–H bond in the adsorption structure. Strong chemical action decreases conductivity, but this effect is minimal in the two Pt atoms doping system.

In the TDOS and PDOS of CH_4_ adsorption, no considerable change is found, as shown in Figure 7b–e. Therefore, the electrical conductivity of the entire system remains unchanged. The 2p orbital of C is mainly distributed between −7.5 and −5 eV. Therefore, the effect on the entire energy band structure is minimal, as demonstrated by the weak physical function and small *E*_ads_ (0.97 and 0.1 eV). Such feature can be used to study new types of sensors with selectivity.

The same analytical method was used to analyze the adsorption of C_2_H_2_. The Pt 5d and C 2p orbitals show simultaneous peaks at the same energy level. The orbital hybridization illustrates the strong interaction between the Pt and C atoms, and the high effect between these atoms also reduces the distribution of TDOS around the Fermi level. Thus, the conductivity of the entire system will also decrease.

Frontier molecular orbital theory was analyzed for different systems, and the effect can be obtained based on the electronic behavior of Pt- and Pt_2_-doped h-BN in the presence of gas molecules. Determining the features that can be practically modified will be helpful for the exploitation of gas sensors. Based on molecular orbital theory, we calculated the HOMO and LUMO distributions of the adsorption system, as shown in Figure 7. We also calculated *E*_g_ to evaluate conductivity changes, as presented in Table 3. HOMO and LUMO are mostly located near the Pt atom doping site, which is associated with the good conductivity and insulation properties of h-BN. After gas adsorption, the LUMO position is drastically changed, whereas the HOMO is slightly altered. As shown in Figure 7a–d, the 1.51 eV *E*_g_ of Pt-h-BN reflects good conductivity. When the gas molecules are close to h-BN, *E*_g_ is rapidly increased to >2.8 eV. As gas molecules transfer charge to the h-BN system, many LUMOs are on the surface of the gas molecules. In the H_2_ molecule model, *E*_g_ can be remarkably improved, and a strong chemical adsorption process occurs, greatly reducing the conductivity of the entire system. Based on the comparison of different situations, almost no HOMO and LUMO distribution occurs on the CH_4_ molecules, and weak physical effects cannot largely change the conductivity. The contribution of C_2_H_2_ to the conductivity is slightly lesser than that of H_2_. Thus, the conductivity of the entire system is slightly higher than that of the H_2_ adsorption system. Figure 7e–h show that the HOMO and LUMO distribution of the gas molecule/Pt_2_-h-BN system presents a similar situation. CH_4_ exhibits a minor role, and *E*_g_ remains unchanged. In general, the molecular adsorption of H_2_ shows the largest influence on conductivity, whereas that of CH_4_ exhibits nearly no effect on conductivity.

In summary, for different adsorption scenarios, conductivity decreases at varying degrees. H_2_ molecular adsorption exhibits the greatest influence on conductivity, followed by C_2_H_2_ adsorption. CH_4_ molecule adsorption is very weak, minimally contributing to conductivity. The estimated final conductivity is arranged as follows: CH_4_ adsorption system >C_2_H_2_ adsorption system >H_2_ adsorption system.

### 3.5. Adsorption of Gas Molecules on Pt_3_, Pt_4_ Doped h-BN

The gas adsorption system on Pt_3_- and Pt_4_-doped h-BN is depicted in Figure 8a–f. Table 4 shows very similar results between Pt- and Pt_2_-doped h-BN systems. Doping structures with more modified Pt atoms are more responsive to H_2_ and C_2_H_2_ than to CH_4_, and the adsorption process with the CH_4_ molecule is very weak. Drastic changes in the charge transfer amount occur at different situations. Different doping structures were compared, and the results are presented as follows.

In the H_2_ adsorption system, a strong chemical adsorption process exists between Pt_3_- or Pt_4_-doped structure and the H_2_ molecule, the adsorption energy slightly decreasing from −1.97 eV to −1.67 eV. Different degrees of adsorption process mechanism cause large changes in *Q*_t_. In the H_2_/Pt_3_-h-BN system, such a large *Q*_t_ indicates that the H atom acquires numerous electrons. Therefore, great changes in conductivity exist.

In the CH_4_ adsorption model, the adsorption distance of approximately 2.30 Å and adsorption energy of <1 eV greatly reflect the weak physical effects, which is consistent with previous conclusions. In the system of C_2_H_2_/Pt_3_- and Pt_4_-doped h-BN, *E*_ads_ of 2.16 and 2.88 eV are approaching that of the C_2_H_2_/Pt- and Pt_2_-doped h-BN systems. This result illustrates that *E*_ads_ remains unchanged, and only slight differences in conductivity occurs.

The TDOS and PDOS of gas molecule adsorption on Pt_3_- and Pt_4_-doped h-BN are presented in Figure 9. When three Pt atoms are doping, TDOS is decreased rapidly at the Fermi level, and the TDOS to the right of the Fermi level is increased, indicating increased electronic filling conduction band. The band gap of Pt_3_-h-BN is 0.01 eV, which reflects very good electrical conductivity. Additional impurity bands and chemical action worsens the conductivity. Figure 9b,c shows a spike at −5 eV, which is caused by the hybridization of the C-2p orbit and Pt-5d orbit from the PDOS. In C_2_H_2_ adsorption, the C-2p and Pt-5d orbits show strong hybridization at many energy levels. Strong molecular forces exhibit a great effect between the two atoms. Thus, the C_2_H_2_ molecule undergoes a great bend.

The overall change of TDOS in the Pt_4_-h-BN adsorption system is relatively weak, as shown in Figure 9d–f. Only the adsorption of the H_2_ gas molecule changes the TDOS at the Fermi level, and a low TDOS decreases the conductivity in this situation. As shown in Figure 9e, almost no change occurs in TDOS around the Fermi level, indicating that the CH_4_ molecule contributes slightly to conductivity. The PDOS from −2.5 eV to 2.5 eV in this case is also a good illustration of this phenomenon. Almost no electronic filling is expected in the Pt atom. Only the Pt doping changes the band structure. Therefore, the conductivity does not change substantially after CH_4_ adsorption. The results in Figure 9f–f1 are almost identical to the previous analysis of C_2_H_2_. Extraordinary orbit hybridizations occur between the C-2p and Pt-5d orbits, but the change in conductivity may be small due to the weak change around the Fermi level.

Based on the molecular orbital theory, the HOMO and LUMO distributions of the adsorbed structures were determined using the same formula. Figure 10 shows an intuitive graphical distribution, and Table 5 lists the arranged data.

As shown in Figure 10, *E*_g_ is decreased between HOMO and LUMO, indicating that increased metal atom doping can reduce *E*_g_, thereby increasing the conductivity. In other respects, the contribution of gas molecules to electrical conductivity is not as remarkable as previously observed, but the degree of discrimination is still quite valuable. In general, gas adsorption can decrease the conductivity of the system, and conductivity caused by Pt_3_ doping is more intense than that by Pt_4_ doping.

In the Pt_3_ doping system, the *E*_g_ of 0.1 eV reflects good electrical conductivity of the system. When accompanied by the adsorption of different gases, the conductivity is decreased, especially when H_2_ is close to the doped structure. *E*_g_ is changed from 0.1 eV to 1.15 eV, indicating that strong adsorption with chemical bond formation decreases electrical conductivity. The role of CH_4_ is non-negligible under this model, and the relatively large *E*_g_ (0.63 eV) also remarkably changes the conductivity of the system. The same effect is observed in the C_2_H_2_ adsorption model. Finally, we analyzed the gas/Pt_4_-h-BN. As shown in Figure 10e–h, the effect of gas molecules on conductivity is slightly decreased, and no considerable changes occur after gas adsorption.

In summary. The adsorption of different molecules decreases conductivity at varying degrees. The adsorption of H_2_ and CH_4_ shows the greatest and weakest influence on conductivity, respectively. The final conductivity is ordered as follows: CH_4_ adsorption system >C_2_H_2_ adsorption system >H_2_ adsorption system, which is the same as that in Pt- and Pt_2_-doped h-BN systems.

### 3.6. Comparison of Charge Transfer, Adsorption Energy, Energy Gap at Different Systems Through Table and Histogram

A comparative analysis of adsorption parameters (*Q_t_*, *E_ads_*, *E_g_*) are carried out to present more intuitive comparison as shown in Table 6 and Figure 11. After gas molecules adsorption, *Q_t_* is ranging from −0.05–0.28 *e*. It is obvious that the charge transfer amount of CH_4_ is smaller than H_2_ and C_2_H_2_. By comparing the *Q_t_* and *E_ads_* of each adsorption systems, chemical effect occurs in in H_2_, C_2_H_2_, while physical effect occurs in CH_4_ during the adsorption process. The adsorption ability for these decomposition components are ordered as follows: C_2_H_2_ > H_2_ > CH_4_. As shown in Figure 11c, *E_g_* has different degrees of reduction after different Pt atoms modification, meaning that the conductivity of the entire system increases. When gas molecules are close to the Pt_n_-h-BN, the rising *E_g_* indicates that the conductivity of the system decrease with different degrees, H_2_ and C_2_H_2_ molecules drastically decrease the conductivity, while CH_4_ nearly does not affect electronic distribution. The final conductivity is ordered as follows: CH_4_ adsorption system >C_2_H_2_ adsorption system >H_2_ adsorption system. This work can be applied to develop new gas sensors based on the Pt cluster modified h-BN using online monitoring of the dissolved gases in transformer oil.

## 4. Conclusions

In this study, the most stable structure of h-BN under various doping strategies is analyzed, and the Pt cluster-modified h-BN is employed to investigate its adsorbing performance on oil-decomposed components. DOS analysis and frontier molecular orbital theory are considered to comprehensively understand the interactions between gas molecules and Pt cluster-modified h-BN. A new gas sensor with selectivity and sensitivity can be developed by analyzing the responsiveness to typical gases. The specific response mechanism and conclusions are as follows:Pt cluster-modified h-BN exhibits good sensitivity to C_2_H_2_ and H_2_ due to chemical adsorption process but is insensitive to CH_4_, with only weak physical adsorption process between them. The adsorption ability for these decomposition components occurs in the following order: C_2_H_2_ > H_2_ > CH_4_.In all doping systems, the adsorption of each gas molecule decreases the conductivity of the entire system. H_2_ and C_2_H_2_ molecules can drastically change the conductivity, and the decreased conductivity values for C_2_H_2_ and H_2_ are ordered as follows: H_2_ > C_2_H_2_. CH_4_ does not affect electronic distribution.In doping by different Pt atoms, the adsorption process mechanisms and adsorption are slightly different, but the good sensitivity to H_2_ and C_2_H_2_ is consistent. On the basis of the large difference in characteristics after adsorption, Pt cluster-modified h-BN is a suitable gas sensor.

Therefore, our calculations suggest that Pt cluster-modified h-BN prepared sensors provide a facile way to practically detect transformer oil-dissolved components and can effectively estimate the operation status of transformers. Pt cluster-modified h-BN is a promising material as a gas sensor and for employment in power systems.

## Figures and Tables

**Figure 1 nanomaterials-09-01746-f001:**
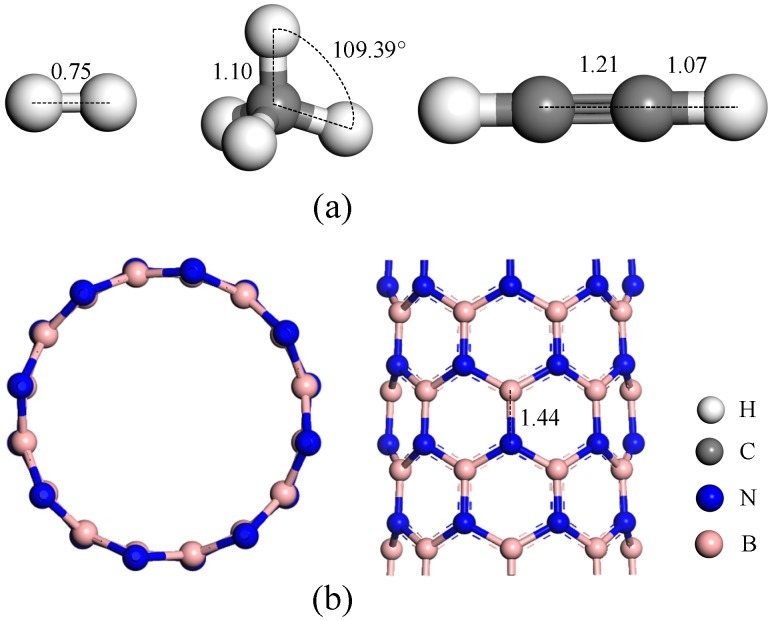
(**a**) H_2_, CH_4_, C_2_H_2_ molecules (**b**) Intrinsic hexagonal boron nitride nanotubes (h-BN).

**Figure 2 nanomaterials-09-01746-f002:**
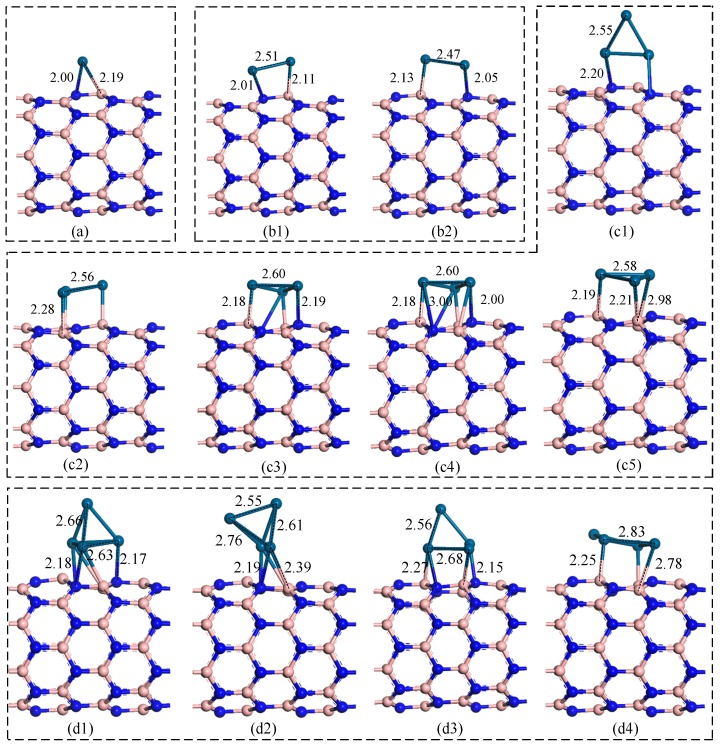
The structures of Pt cluster modified h-BN. The h-BN is modified by a cluster containing a Pt atom (**a**), two Pt atom (**b1**,**b2**), three Pt atom (**c1**–**c5**), four Pt atom (**d1**–**d4**).

**Figure 3 nanomaterials-09-01746-f003:**
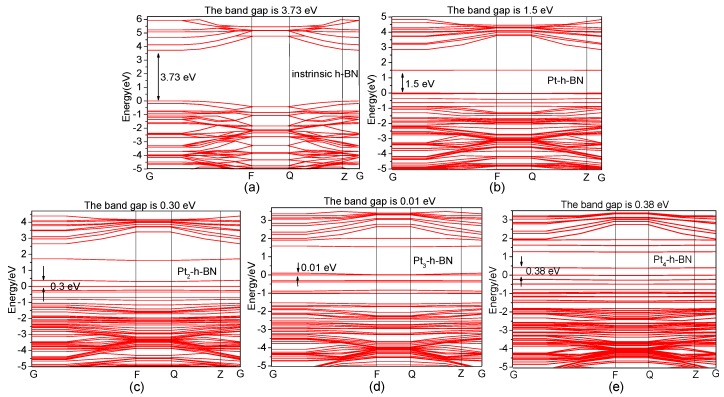
Energy band diagram of different structures, (**a**) intrinsic h-BN, (**b**) Pt-h-BN, (**c**) Pt_2_-h-BN, (**d**) Pt_3_-h-BN, (**e**) Pt_4_-h-BN.

**Figure 4 nanomaterials-09-01746-f004:**
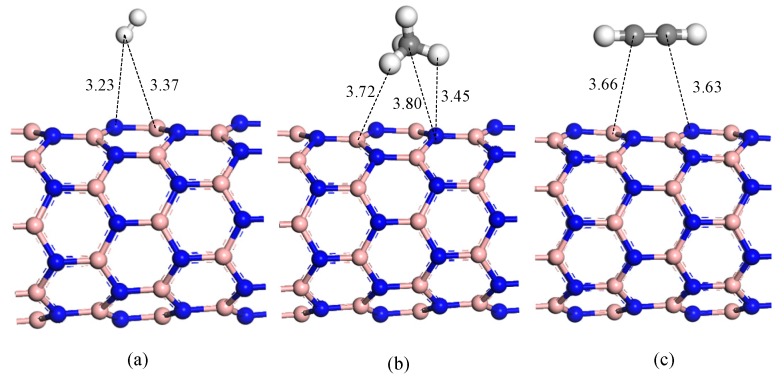
The structures of gas molecular adsorption on intrinsic h-BN. (**a**) H_2_, (**b**) CH_4_, (**c**) C_2_H_2_.

**Figure 5 nanomaterials-09-01746-f005:**
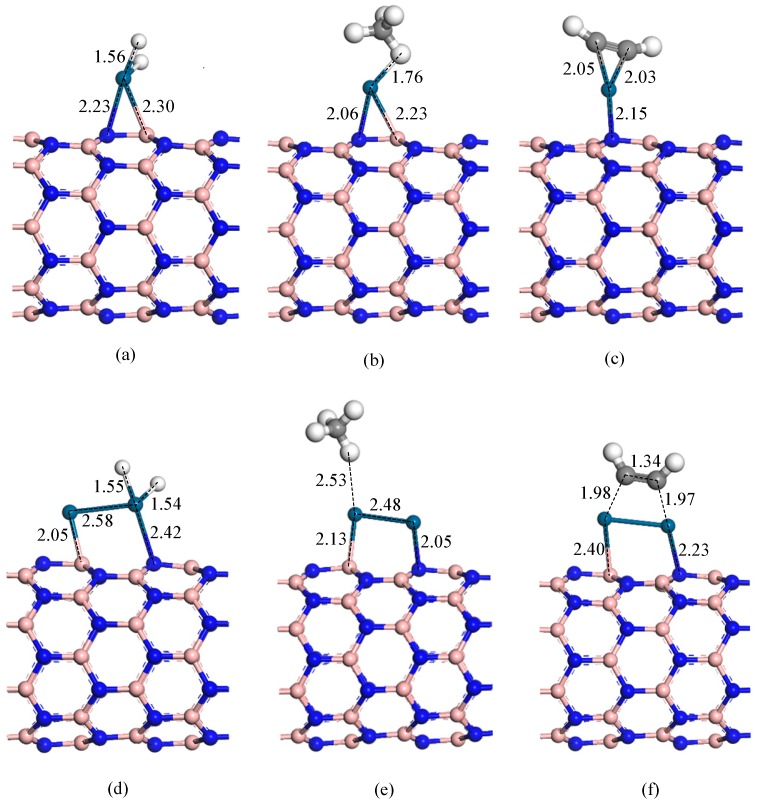
Adsorption of gas molecules on Pt, Pt_2_ modified h-BN. (**a**) H_2_ molecule on Pt modified h-BN, (**b**) CH_4_ molecule on Pt modified h-BN, (**c**) C_2_H_2_ molecule on Pt modified h-BN, (**d**) H_2_ molecule on Pt_2_ modified h-BN, (**e**) CH_4_ molecule on Pt_2_ modified h-BN, (**f**) C_2_H_2_ molecule on Pt_2_ modified h-BN.

**Figure 6 nanomaterials-09-01746-f006:**
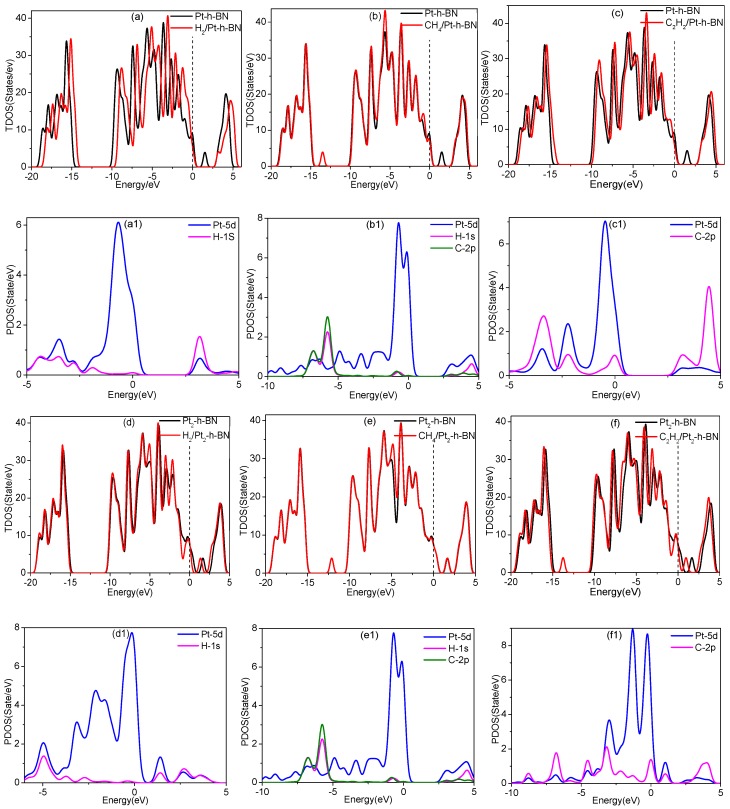
Total density of states (TDOS) and partial density of states (PDOS) of adsorption of gas molecules on Pt, Pt_2_ modified h-BN. (**a**) TDOS of H_2_ on Pt modified h-BN, (**b**) TDOS of CH_4_ on Pt modified h-BN, (**c**) TDOS of C_2_H_2_ on Pt modified h-BN, (**d**) TDOS of H_2_ on Pt_2_ modified h-BN, (**e**) TDOS of CH4 on Pt_2_ modified h-BN, (**f**) TDOS of C2H2 on Pt_2_ modified h-BN. (**a1**) to (**f1**) are PDOS of the corresponding subgraph.

**Figure 7 nanomaterials-09-01746-f007:**
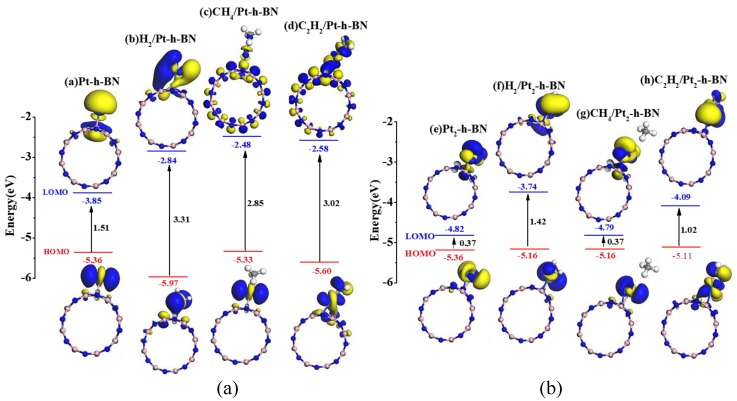
Molecular orbital before and after gas adsorption. (**a**) Pt-h-BN system, (**b**) Pt_2_-h-BN system.

**Figure 8 nanomaterials-09-01746-f008:**
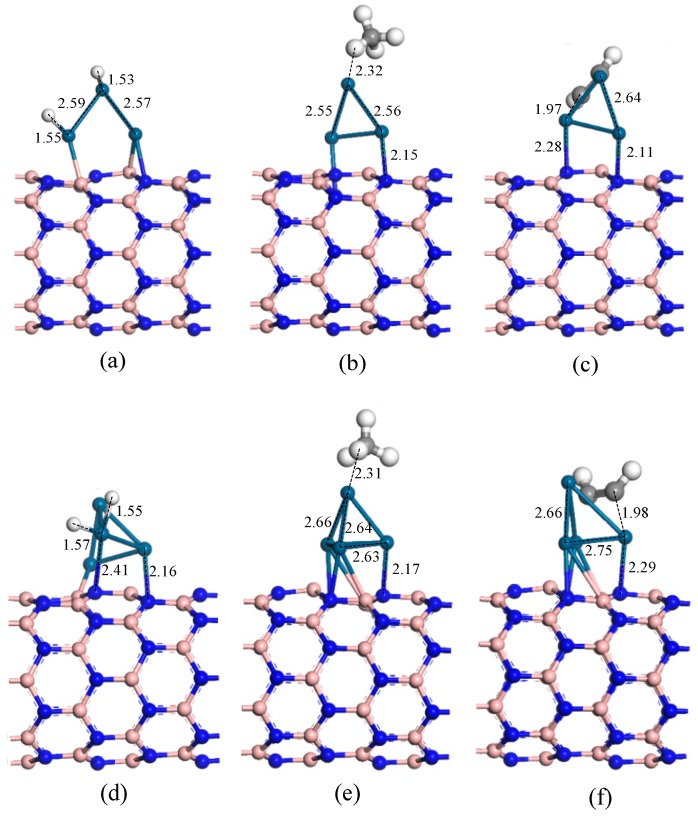
Adsorption of gas molecules on Pt_3_, Pt_4_ doped h-BN. (**a**) H_2_ molecule on Pt_3_ modified h-BN, (**b**) CH_4_ molecule on Pt_3_ modified h-BN, (**c**) C_2_H_2_ molecule on Pt_3_ modified h-BN, (**d**) H_2_ molecule on Pt_4_ modified h-BN, (**e**) CH_4_ molecule on Pt_4_ modified h-BN, (**f**) C_2_H_2_ molecule on Pt4 modified h-BN.

**Figure 9 nanomaterials-09-01746-f009:**
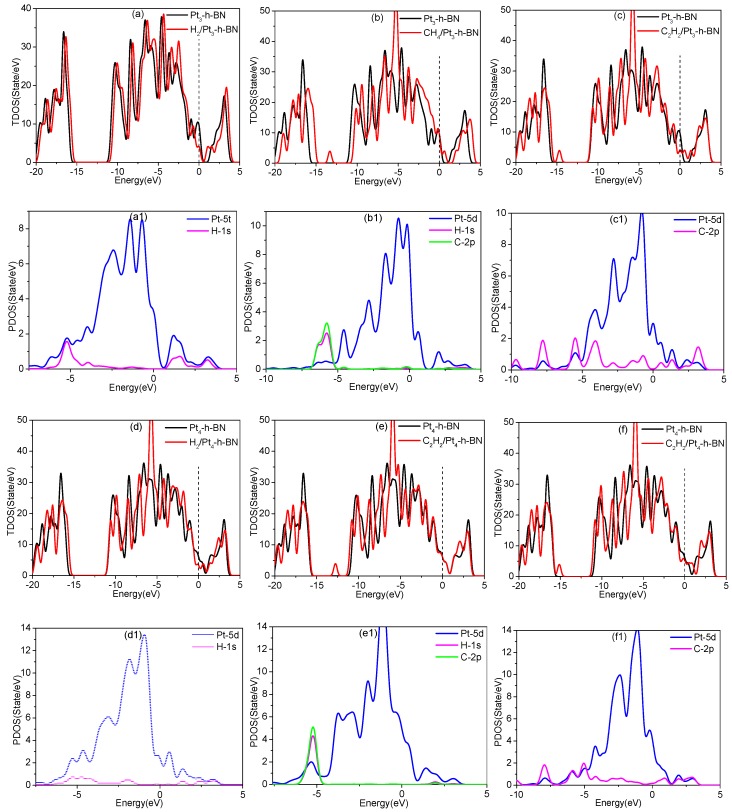
TDOS and PDOS of adsorption of gas molecules on Pt_3_, Pt_4_ modified h-BN. (**a**) TDOS of H_2_ on Pt_3_ modified h-BN, (**b**) TDOS of CH_4_ on Pt_3_ modified h-BN, (**c**) TDOS of C_2_H_2_ on Pt_3_ modified h-BN, (**d**) TDOS of H_2_ on Pt_4_ modified h-BN, (**e**) TDOS of CH4 on Pt_4_ modified h-BN, (**f**) TDOS of C_2_H_2_ on Pt_4_ modified h-BN. (**a1**) to (**f1**) are PDOS of the corresponding subgraph.

**Figure 10 nanomaterials-09-01746-f010:**
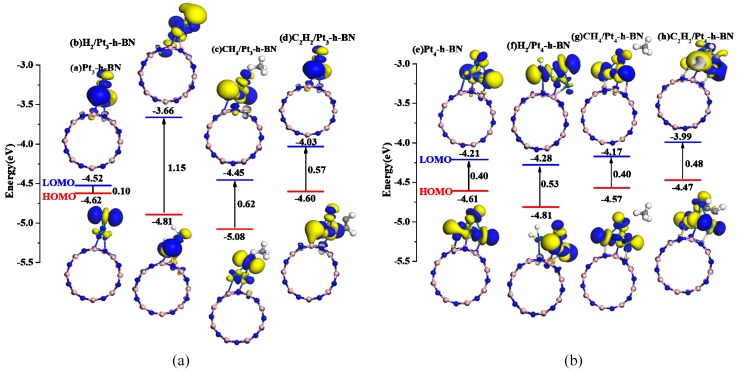
Molecular orbital before and after gas adsorption. (**a**) Pt_3_-h-BN system, (**b**) Pt_4_-h-BN system.

**Figure 11 nanomaterials-09-01746-f011:**
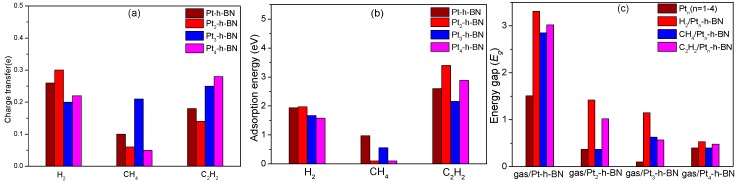
Contrast histogram analysis of adsorption parameters. (**a**) Charge transfer of all adsorption system, (**b**) adsorption of all adsorption system, (**c**) energy gap of all adsorption system.

**Table 1 nanomaterials-09-01746-t001:** Adsorption process parameters of adsorption of H_2_, CH_4_, and C_2_H_2_ on the surface of intrinsic h-BN.

System	Structure	Adsorption Distance (d)	Adsorption Energy (*E*_ads_)
H_2_ adsorption	Figure 4a	3.23 Å	−0.08 eV
CH_4_ adsorption	Figure 4b	3.45 Å	−0.06 eV
C_2_H_2_ adsorption	Figure 4c	3.63 Å	−0.10 eV

**Table 2 nanomaterials-09-01746-t002:** Adsorption parameters of gas molecules on Pt, Pt_2_ doped h-BN: charge transfer (*Q*_t_), adsorption distance (d), adsorption energy (*E*_ads_).

System	Structure	Adsorption Distance (6)	Adsorption Energy (*E*_ads_)	Charge Transfer (*Q*_t_)
H_2_/Pt-h-BN	Figure 5a	1.56 Å	−1.94 eV	0.26 *e*
CH_4_/Pt-h-BN	Figure 5b	1.76 Å	−0.97 eV	0.10 *e*
C_2_H_2_/Pt-h-BN	Figure 5c	2.03 Å	−2.60 eV	0.18 *e*
H_2_/Pt_2_-h-BN	Figure 5d	1.54 Å	−1.97 eV	0.30 *e*
CH_4_/Pt_2_-h-BN	Figure 5e	2.53 Å	−0.10 eV	0.06 *e*
C_2_H_2_/Pt_2_-h-BN	Figure 5f	1.97 Å	−3.40 eV	0.14 *e*

**Table 3 nanomaterials-09-01746-t003:** Molecular orbitals and energy gaps of individual molecules before and after adsorption.

Adsorption system	Structure	*E*_HOMO_(eV)	*E*_LUMO_(eV)	*E*_g_(eV)
Pt-h-BN	Figure 7a	−5.34	−3.85	1.51
H_2_/Pt-h-BN	Figure 7b	−5.97	−2.84	3.31
CH_4_/Pt-h-BN	Figure 7c	−5.33	−2.48	2.85
C_2_H_2_/Pt-h-BN	Figure 7d	−5.60	−2.58	3.02
Pt_2_-h-BN	Figure 7e	−5.19	−4.82	0.37
H_2_/Pt_2_-h-BN	Figure 7f	−5.16	−3.74	1.42
CH_4_/Pt_2_-h-BN	Figure 7g	−5.16	−4.79	0.37
C_2_H_2_/Pt_2_-h-BN	Figure 7h	−5.11	−4.09	1.02

**Table 4 nanomaterials-09-01746-t004:** Adsorption parameters of gas molecules on Pt_3_, Pt_4_ doped h-BN: charge transfer (*Q*_t_), adsorption distance (**d**), adsorption energy (*E*_ads_).

System	Structure	Adsorption Distance (d)	Adsorption Energy (*E*_ads_)	Charge Transfer (*Q*_t_)
H_2_/Pt_3_-h-BN	Figure 8a	1.53 Å	1.67 eV	0.20 *e*
CH_4_/Pt_3_-h-BN	Figure 8b	2.32 Å	0.56 eV	0.21 *e*
C_2_H_2_/Pt_3_-h-BN	Figure 8c	1.97 Å	2.16 eV	0.25 *e*
H_2_/Pt_4_-h-BN	Figure 8d	1.55 Å	1.58 eV	0.22 *e*
CH_4_/Pt_4_-h-BN	Figure 8e	2.31 Å	0.10 eV	0.05 *e*
C_2_H_2_/Pt_4_-h-BN	Figure 8f	1.98 Å	2.88 eV	0.28 *e*

**Table 5 nanomaterials-09-01746-t005:** Molecular orbitals and energy gaps of individual molecules before and after adsorption.

Adsorption System	Structure	*E*_HOMO_ (eV)	*E*_LUMO_ (eV)	*E*_g_ (eV)
Pt_3_-h-BN	Figure 10a	−4.62	−4.52	0.10
H_2_/Pt_3_-h-BN	Figure 10b	−4.81	−3.66	1.15
CH_4_/Pt_3_-h-BN	Figure 10c	−5.08	−4.45	0.63
C_2_H_2_/Pt_3_-h-BN	Figure 10d	−4.60	−4.03	0.57
Pt_4_-h-BN	Figure 10e	−4.61	−4.21	0.40
H_2_/Pt_4_-h-BN	Figure 10f	−4.81	−4.28	0.53
CH_4_/Pt_4_-h-BN	Figure 10g	−4.57	−4.17	0.40
C_2_H_2_/Pt_4_-h-BN	Figure 10h	−4.47	−3.99	0.48

**Table 6 nanomaterials-09-01746-t006:** Adsorption parameters summary of different systems.

Adsorption Type	*Q_t_* (*e*)	*E_ads_* (eV)	*E_g_* (eV)
Pt-h-BN	/	/	1.51
Pt_2_-H-BN	/	/	0.37
Pt_3_-h-BN	/	/	0.10
Pt_4_-h-BN	/	/	0.40
H_2_/Pt-h-BN	0.26	1.94	3.31
H_2_/Pt_2_-h-BN	0.30	1.97	1.42
H_2_/Pt_3_-h-BN	0.20	1.67	1.15
H_2_/Pt_4_-h-BN	0.22	1.58	0.53
CH_4_/Pt-h-BN	0.10	0.97	2.85
CH_4_/Pt_2_-h-BN	0.06	0.10	0.37
CH_4_/Pt_3_-h-BN	0.21	0.56	0.63
CH_4_/Pt_4_-h-BN	0.05	0.10	0.40
C_2_H_2_/Pt-h-BN	0.18	2.60	3.02
C_2_H_2_/Pt_2_-h-BN	0.14	3.40	1.02
C_2_H_2_/Pt_3_-h-BN	0.25	2.16	0.57
C_2_H_2_/Pt_4_-h-BN	0.28	2.88	0.48

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
