# Peer review of "Pt Cluster Modified h-BN for Gas Sensing and Adsorption of Dissolved Gases in Transformer Oil: A Density Functional Theory Study"

_nanomaterials, 2019, doi:10.3390/nano9121746_

Round 1
Reviewer 1 Report
The manuscript entitled: " Pt cluster modified h-BN for gas sensing and adsorption of dissolved gases in transformer oil: A DFT study", has an interesting scientific approach on a very actual subject and is clearly written.
Nevertheless, some aspects need to be revised.
1. As first time mentioning, please introduce in the Abstract Boron nitride nanotubes (BN), before abbreviation.
2. Abstract, line 13, "Pt cluster-modified h-BN 12 is proposed a novel gas sensor"
has to be changed in Pt cluster-modified h-BN 12 is proposed as a sensitive material for a novel gas sensor
3. In each case replace molecular orbit with molecular orbital. This recognition process is not based on chemical reaction or bonds, but on physical phenomena
4. Line 120, replace table with correct most stable structures
5. Please replace "The H2, C2H2, and CH4 molecular structures remain unchanged during the reaction, with The H2, C2H2, and
CH4 molecular structures remain unchanged during the adsorption process,
6. In each case replace reaction with adsorption process
7. Physical reactions (line 200) in this case is an inadequate expression.
8. Line 257, please reformulate in an adequate and clear way "and numerous LUMOs are transferred to gas molecules"
9. References 7, 18, 20 are not complete.
Author Response
Point 1: As first time mentioning, please introduce in the Abstract Boron nitride nanotubes (BN), before abbreviation.
Response 1: We have added the full name before the abbreviation.
Point 2: Abstract, line 13, "Pt cluster-modified h-BN 12 is proposed a novel gas sensor"
has to be changed in Pt cluster-modified h-BN 12 is proposed as a sensitive material for a novel gas sensor
Response 2: The statement has been edited as you suggested.
Point 3: In each case replace molecular orbit with molecular orbital. This recognition process is not based on chemical reaction or bonds, but on physical phenomena.
Response 3: The words has been replaced as you suggested.
Point 4: Line 120, replace table with correct most stable structures
Response 4: The statement has been replaced as you suggested.
5. Please replace "The H2, C2H2, and CH4 molecular structures remain unchanged during the reaction, with The H2, C2H2, and
CH4 molecular structures remain unchanged during the adsorption process,
Response 5: The statement has been replaced as you suggested.
6. In each case replace reaction with adsorption process
Response 6: We have done the replacement.
7. Physical reactions (line 200) in this case is an inadequate expression.
Response 7: This statement has been corrected to physical adsorption.
8. Line 257, please reformulate in an adequate and clear way "and numerous LUMOs are transferred to gas molecules"
Response 8: This statement has been changed to be more appropriate and clear.
9. References 7, 18, 20 are not complete.
Response 9: We have completed the information about the reference you mentined and reference 27 has also been corrected correctly.
Reviewer 2 Report
It is a very interesting paper describing theoretical calculations of adsorption behavior of Ptn-h-BN to different gases such as C2H2, H2 and CH4 . A number of publications have demonstrated, CH4 adsorption on boron nitride and also on Pt. It is not clear why Ptn-h-BN shoud behave differently in the adsorption of C2H2 and CH4.If the behavior is due to structural differences
between C2H2 (linear) to tetrahedral structure of CH4 as described, these structural differences had not operated in BN or Pt as both the hydrocarbons are adsorbed on both the surfaces. As such the parameters used in the DFT calculations need to be examined and modified.
M.D. Ganji, A. Mirnejad, and A. Najafi, Sci Technol Adv Mater. 2010 Aug; 11(4): 045001.
Y. Wang, L. Zhao, L.Shi, J. Sheng, W. Zhang, X. Cao, P. Hu and L. An-Hui, Catal. Sci. Technol., 2018, 8, 2051–2055.
Author Response
Point 1: A number of publications have demonstrated, CH4 adsorption on boron nitride and also on Pt. It is not clear why Ptn-h-BN should behave differently in the adsorption of C2H2 and CH4. If the behavior is due to structural differences between C2H2 (linear) to tetrahedral structure of CH4 as described, these structural differences had not operated in BN or Pt as both the hydrocarbons are adsorbed on both the surfaces. As such the parameters used in the DFT calculations need to be examined and modified.
M.D. Ganji, A. Mirnejad, and A. Najafi, Sci Technol Adv Mater. 2010 Aug; 11(4): 045001.
Y. Wang, L. Zhao, L.Shi, J. Sheng, W. Zhang, X. Cao, P. Hu and L. An-Hui, Catal. Sci. Technol., 2018, 8, 2051–2055.
Response 1: Thanks for your advice. We have read the references you mentioned, and we think that our calculation parameters may not need to be modified.
In the first reference you mentioned, the binding energy between boron nitride nanotubes (8, 0) and CH4 was -2.79 kcal mol−1 (-0.12 eV), and concluded: “Thus, we conclude that BNNTs might be unsuitable for the storage of methane and thus of natural gases.”, which is consistent with our work. It is difficult for CH4 to adsorb on boron nitride nanotubes (BNNTs) under normal conditions. Meanwhile, the adsorption is enhanced with the addition of Ptn in our work, and this is consistent with common sense.
In the second reference you mentioned, they proposed boron nitride (BN) as a catalyst for CH4 and O2 reactions. But there is a difference between the BN they studied and the BNNTs in our work, the BN is a 2D material with graphene-like structure. In addition, BN as the catalyst requires a higher reaction temperature as the authors said: “Furthermore, the first C–H bond dissociation of CH4 has to overcome the energy barrier as high as 2.29 eV, and the reaction energy of this step is evidently endothermic. Hence, it is formidable for the B–O–B sites to activate either CH4 or O2 even at ∼700°C.”.
Reviewer 3 Report
In this study, the authors reported results on the most stable structure of h-BN under doping by various number of Pt atoms. The authors investigated the sensitivity and selectivity of considered doped h-BN materials toward the detection of different gas C2H2, H2 and CH4. To investigate the interactions between gas molecules and Pt cluster-modified h-BN, the authors used DOS analysis and frontier molecular orbital theory. They demonstrated that Pt cluster-modified h-BN exhibits good sensitivity to C2H2 and H2 but insensitive to CH4. Furthermore, they demonstrated that the adsorption of gas molecule decreases the conductivity of the prepared system, whereas the CH4 does not affect its electronic properties.
The author’s results and discussion could provide a valuable contribution to this research field. These results could be also interesting from the application point of view in developing new device for molecular detection. Herein some comments to improve the manuscript:
The authors must explain how the adsorption distance is determined; it is not clear from the text. The authors must explain how the sensitivity is evaluated from the results; it is not clear from the manuscript The manuscript need little editing.
The present work could be accepted for publication by considering the mentioned points above.
Author Response
Point 1: The authors must explain how the adsorption distance is determined; it is not clear from the text. The authors must explain how the sensitivity is evaluated from the results; it is not clear from the manuscript The manuscript need little editing.

Response 1: Thanks for your advices. We have added an explanation for determining the adsorption distance in line 162, and explained how the sensitivity is evaluated from the results in line 96.